# MiRNA Profiling of Areca Nut-Induced Carcinogenesis in Head and Neck Cancer

**DOI:** 10.3390/cancers16213710

**Published:** 2024-11-03

**Authors:** Hung-Han Huang, Joseph T. Chang, Guo-Rung You, Yu-Fang Fu, Eric Yi-Liang Shen, Yi-Fang Huang, Chia-Rui Shen, Ann-Joy Cheng

**Affiliations:** 1Graduate Institute of Biomedical Sciences, College of Medicine, Chang Gung University, Taoyuan 33302, Taiwan; d1001402@cgu.edu.tw (H.-H.H.); crshen@mail.cgu.edu.tw (C.-R.S.); 2Department of Medical Biotechnology and Laboratory Science, College of Medicine, Chang Gung University, Taoyuan 33302, Taiwan; d000017007@cgu.edu.tw (G.-R.Y.); yufang1998@cgmh.org.tw (Y.-F.F.); 3Department of Radiation Oncology and Proton Therapy Center, Linkou Chang Gung Memorial Hospital, Taoyuan 333423, Taiwan; jtchang@cgmh.org.tw (J.T.C.); pts@cgmh.org.tw (E.Y.-L.S.); 4School of Medicine, Chang Gung University, Taoyuan 33302, Taiwan; 5Department of General Dentistry, Linkou Chang Gung Memorial Hospital, Taoyuan 333423, Taiwan; yifang0324@gmail.com; 6Graduate Institute of Dental and Craniofacial Science, College of Medicine, Chang Gung University, Taoyuan 33302, Taiwan

**Keywords:** head and neck cancer (HNC), areca nut, miRNA profiling, carcinogenesis, miR-499a-5p

## Abstract

This study investigates the critical yet understudied role of miRNAs in areca nut-induced carcinogenesis in head and neck cancer (HNC). Through comprehensive profiling, we identified 84 miRNAs, comprising 39 oncogenic miRNAs (OncomiRs) and 45 tumor-suppressive miRNAs (TsmiRs) associated with areca nut-induced HNC. Further analysis of the oncogenic mechanisms revealed that 740 genes are cross-regulated by a cluster of eight hub TsmiRs. These areca nut-modulated miRNAs significantly impact key cancer-related pathways, including p53, PI3K-AKT, MAPK, and Ras signaling, and regulate critical oncogenic processes related to cell motility and survival. Moreover, miR-499a-5p was validated as a vital regulator, which mitigated areca nut-induced cancer progression, including cell migration, invasion, and chemoresistance. Our findings highlight the potential of this miRNA panel for clinical applications in risk assessment, diagnosis, and prognosis of areca nut-associated malignancies, opening new avenues for future research and clinical practice.

## 1. Introduction

Head and neck cancer (HNC) primarily comprises squamous cell carcinomas affecting the oral cavity, pharynx, larynx, and salivary glands, ranking among the top ten most prevalent cancers worldwide [1]. The prognosis of HNC is influenced by various factors, including the stage of cancer and its subsites [2]. Several established risk factors contribute to the carcinogenesis of HNC, notably cigarette smoking, smokeless tobacco use [3], heavy alcohol consumption [4], and areca nut chewing [5]. Areca nut, particularly prevalent in Southeast Asia, stands out as the most commonly used carcinogen, exhibiting the highest relative risk and contributing to the region’s elevated incidence and mortality rates [6,7]. When combined with other risks, areca nut significantly increases the overall risk, underscoring its profound influence on oral cancer in Southeast Asia [8].

Areca nut, also known as betel nut, is the seed of the *Areca catechu* L. Areca nut extract (ANE), and arecoline, a major alkaloid component, has been demonstrated to induce the pathogenesis of oral cancer and oral submucous fibrosis [9,10]. ANE and arecoline promote cell motility through several mechanisms, including upregulation of matrix metalloproteinases and epithelial–mesenchymal transition (EMT) [11,12]. Moreover, chronic exposure to ANE facilitates cancer stemness conversion, leading to higher resistance to cellular stress, such as chemo-radiotherapy [11,13]. Consequently, HNC patients with a history of areca nut chewing often experience more aggressive disease progression and lower survival rates [14,15]. Understanding the molecular mechanisms triggered by areca nut is urgent, as it could lead to more effective prevention and treatment strategies.

Over the past decade, miRNA research has firmly established this molecular family as a critical cellular process regulator. MiRNAs are a small noncoding RNA (ncRNA) typically containing 18–22 nucleotides that target the 3′-UTR of mRNA, leading to mRNA degradation or inhibition of protein translation [16]. The miRNA family plays essential roles in cell differentiation, growth, and apoptosis, and its dysregulation may contribute to numerous human disorders, including cancer. Recently, miRNA expression has been screened in HNC, with several molecules commonly identified across multiple studies, such as miR-21, miR-31, and miR-196 [17,18,19]. However, knowledge of miRNAs in areca nut-induced molecular carcinogenesis remains limited. Thus far, only one study has profiled miRNA’s response to areca nut and identified upregulation of miR-23a [20]. However, this research was conducted using oral fibroblast cells treated with acute DNA-damaging doses, which does not accurately replicate the conditions of HNC patients with long-term, habitual areca nut exposure.

This study employs a systematic strategy to identify a miRNA panel associated with areca nut-induced HNC. We screened 462 common miRNAs in HNC cells responsive to chronic ANE treatment and assessed the clinical relevance of these miRNAs using TCGA-HNSC dataset. Through integrated analysis of both datasets, we identified a panel of miRNAs implicated in areca nut-induced HNC and explored molecular pathways through which these miRNAs exert their oncogenic effects. One critical molecule, miR-499a-5p, was further investigated to demonstrate its role in areca nut-induced malignant phenotypes. Our findings provide valuable pathological insights into areca nut-induced malignancy that hold the potential to improve patient outcomes in the future.

## 2. Materials and Methods

### 2.1. Cell Culture and Arecoline Treatment

Two normal keratinocytes (NOK and CGHNK2) and three HNC cell lines (OECM1, SAS, and CGHNC9) were used and cultured as previously described [21,22]. Briefly, NOK and CGHNK2 were maintained in keratinocyte serum-free medium (KSFM) supplemented with 50 µg/mL bovine pituitary extract (BPE) and 5 ng/mL epidermal growth factor (EGF) (Thermo Fisher Scientific, Waltham, MA, USA). OECM1 was cultured in RPMI1640 medium (Thermo Fisher Scientific). SAS and CGHNC9 were grown in DMEM medium (Thermo Fisher Scientific). All media for cancer cell lines were supplemented with 10% fetal bovine serum (FBS) and 1% antibiotic antimycotic (Thermo Fisher Scientific).

To establish isogenic sublines of HNC cells with chronic areca nut exposure, the OECM1 and SAS cell lines were chronically treated with an IC30 dose (30% maximal inhibitory concentration) of ANE in a complete culture medium for 3 months, as previously described [21,22]. To determine the effect of miRNA expression and cellular response to areca nut exposure, HNC cells were treated with 100 µM arecoline in a complete medium for 24 h, followed by molecular and cellular analyses.

### 2.2. Screening of miRNAs Using miRNA Microarray

Two HNC cell lines (OECM1, SAS) and their respective sublines chronically treated with ANE were used for miRNA screening. Briefly, cellular RNA was extracted using TRIzol reagent (Life Technologies, Carlsbad, CA, USA) and quantified using the Nanovue™ spectrophotometer (GE Healthcare, Chicago, IL, USA) as previously described [23]. The Human miRNA Microarray Kit (G4470A, Agilent Technologies, Santa Clara, CA, USA), comprising probes for 462 miRNAs from the miRBase database, was used for miRNA screening. Feature Extraction software (Version 10.5.1, Agilent, Santa Clara, CA, USA) was used to extract and convert the intensity signal to miRNA expression. GeneSpring GX software (Version 7.3.1, Agilent, Santa Clara, CA, USA) was employed to analyze miRNA expression data from each chip, calibrated by batch removal to identify differential expression levels of each miRNA.

### 2.3. Determination of miR-499a-5p and Target Gene Expression by RT-qPCR Method

The expression level of miR-499a-5p and target gene expression were analyzed using RT-qPCR. Briefly, total RNA was extracted using TRIzol reagent (Invitrogen, Carlsbad, CA, USA) according to manufacturer’s protocol. For miR-499a-5p analysis, reverse transcription and PCR were performed using TaqMan™ MicroRNA Assay kits (Thermo Fisher Scientific). The expression of miR-499a-5p (Assay ID #001352) was evaluated using iQ^TM^ supermix (Bio-rad, Hercules, CA, USA) with RNU6B (Assay ID #001093) serving as an internal control. All miR-499a-5p analyses were conducted following the TaqMan Small RNA Assays user guide (Thermo Fisher Scientific). For target gene expression analysis, quantification was performed using iQ SYBR Green Supermix (Bio-rad), with GAPDH as an internal control. All primer sequences used for target gene amplification were provided in Appendix A.

### 2.4. Overexpression of miR-499a-5p via miRNA-Mimics Transfection

The miR-499a-5p mimic was constructed using pair-specific oligonucleotides: sense 5′-UUA AGA CUU GCA GUG AUG UUU-3′ and antisense 5′-AAA CAU CAC UGC AAG UCU UAA-3′, designed and synthesized by GenDiscovery Biotechnology, Inc. (NTC, Taiwan). To overexpress miR-499a-5p, 300 pmol of the mimics were transfected into OECM1 or SAS cells using Jetprime^®^ transfection reagent (Polyplus, Illkirch, France) or Lipofectamine 2000 (Invitrogen, Carlsbad, CA, USA) according to the manufacturer’s instructions. Molecular and cellular effects were assessed 24 h post-transfection. RT-qPCR was performed to validate the efficacy of miR-499a-5p transfection.

### 2.5. Analysis of Cell Invasion Ability via Matrigel Invasion Assay

Cell invasion ability was evaluated using a BioCoat Matrigel invasion assay, as previously described [24,25]. Briefly, 5% Matrigel (Becton Dickinson Biosciences, Bedford, MA, USA) was coated onto the membrane of the upper insert of the Millicell invasion chamber (Millipore, Burlington, MA) with an 8 µm pore size in a 24-well plate. Cells in serum-free medium were seeded in the upper chamber, while the bottom chamber contained medium with 20% FBS to trap the invading cells. After 24 h, the invasive cells on the reverse side of the upper insert were fixed and stained with crystal violet for observation. ImageJ software (National Institutes of Health, Bethesda, MD, USA) was used to quantify the area of crystal violet stain to determine the invasion ability of each cellular group [26].

### 2.6. Analysis of Cell Migration Ability via In Vitro Wound Healing Assay

Cell migration was examined by an in vitro wound healing assay, as previously described [24,25]. Briefly, cells were seeded in ibidi culture inserts (ibidi, Gräfelfing, Germany) on top of a six-well plate. After eight hours of incubation, the culture inserts were detached to form a cell-free gap in a monolayer of cells. After changing to serum-free media, the cell migration status toward the gap area was photographed at intervals. The coverage area of the cellular gap was quantified by ImageJ software.

### 2.7. Assessment of Chemosensitivity by Cell Survival Assay

Chemosensitivity was determined by a cell survival assay after cisplatin treatment. Briefly, 2500 cells in 100 µL medium were seeded in a 96-well plate. Once the cells adhered, various concentrations of cisplatin (0–5 µg/mL) were added to each well. After 48 h, Cell Counting Kit-8 (CCK8) reagent was added to each well to determine cell viability according to the manufacturer’s instructions (Dojindo, Kumamoto, Japan). The survival fraction was calculated as the ratio of cell viability in the cisplatin-treated group compared to the untreated group.

### 2.8. TCGA-HNSC Public Data, Bioinformatics Algorithms, and Pathway Analysis

Public miRNA and mRNA expression data from TCGA-HNSC dataset were retrieved from the University of California, Santa Cruz (UCSC) Cancer Genome Browser [27]. This clinical dataset contained 477 HNSC tissues and 44 adjacent normal tissues. To calculate tumor purity in each patient sample, the Estimation of STromal and Immune cells in MAlignant Tumors using Expression data (ESTIMATE) tool was used [28]. Significant differences in miRNA expression between tumor and normal tissues were calculated using an unpaired *t*-test.

To identify miRNA targets, the R package “multiMiR” (Version 1.28.0) was applied [29]. This package includes 11 databases with three based on experimental validation (miRecords, miRTarBase, and TarBase) and eight based on prediction (DIANA-microT-CDS, ElMMo, MicroCosm, miRanda, miRDB, PicTar, PITA, and TargetScan). For functional pathway analysis, the Kyoto Encyclopedia of Genes and Genomes (KEGG) pathway analytical tool was utilized through The Database for Annotation, Visualization, and Integrated Discovery (DAVID) [30]. The interaction network between miR-499a-5p and its target genes was visualized using Cytoscape software (Version 3.9.1).

## 3. Results

### 3.1. Identification of a miRNA Panel Induced by Areca Nut in HNC Cells

To explore potential miRNAs responsive to areca nut exposure in HNC cells, we screened miRNA profiles in two HNC cell lines, OECM1 and SAS, both chronically treated with ANE (ANE sublines). A miRNA microarray analysis was performed to examine the differential expressions in two paired HNC cell lines: the parental (PT) cells and the ANE sublines. Using a dysregulation threshold of absolute fold-regulation ≥ 1.2, we identified 256 dysregulated miRNAs in OECM1 cells and 250 dysregulated miRNAs in SAS cells (Figure 1A, Appendix A). The most significantly dysregulated miRNAs affected by ANE, including miR-513a-5p and miR-548c-3p, are shown in Figure 1B. To compare miRNA expression between OECM1 and SAS cell lines, we categorized miRNAs into eight groups based on their expression patterns (Figure 1C). In total, 292 miRNAs were dysregulated, with 136 miRNAs upregulated (R1 + R2 + R3) and 156 miRNAs downregulated (R5 + R6 + R7) in at least one cell line. These results highlight the potential molecular alterations in response to ANE exposure. Focusing on the most significant miRNAs induced by areca nut, we identified 118 miRNAs commonly altered in both ANE sublines, with 60 upregulated (R2) and 58 downregulated (R6) (Figure 1D, Appendix A). Consequently, we defined a panel of 292 miRNAs responsive to areca nut exposure in HNC cells, with 118 of these being the most significantly altered in both cell lines.

### 3.2. Identification of a miRNA Panel Relevant to HNC Development

To identify miRNAs related to development of HNC, we analyzed the differential expression levels of miRNAs between tumors (*n* = 477) and adjacent normal tissues (*n* = 44) from HNC patients using TCGA-HNSC dataset. We used the ESTIMATE package (Version 1.0.13) to acquire miRNA expression levels in the tumors, as this algorithm can precisely calculate tumor purity and calibrate miRNA levels accordingly. Using selection criteria of average absolute fold-regulation ≥ 1.2 and *p* < 0.05 between normal and tumor groups, we identified 692 differentially expressed miRNAs, with 449 overexpressed and 243 underexpressed in tumor tissues (Figure 2A, Appendix A). Examples of overexpressed miRNAs included miR-105-5p, miR-767-5p, and miR-196b-5p, while underexpressed miRNAs included miR-1-3p, miR-451a, and miR-30a-5p (Figure 2B). This analysis defined a panel of miRNAs dysregulated in HNC patient tumors.

### 3.3. Identification of miRNA Signatures Induced by Areca Nut and Contributing to HNC

To explore miRNAs that contribute to areca nut-induced HNC, we conducted an integrative analysis of two miRNA panels: those induced by ANE and those relevant to the HNC development. For the oncogenic miRNA (OncomiR) set, we combined the analysis of 136 upregulated miRNAs in ANE sublines and 449 overexpressed miRNAs in tumor tissues. This analysis identified 39 miRNAs (Figure 3A, Table 1), revealing a cluster of areca nut-induced OncomiRs involved in HNC development. To identify the most significant OncomiRs, we plotted these miRNAs according to their levels of ANE induction and tumor overexpression. As shown in Figure 3B, ten OncomiRs exhibiting high levels (absolute fold-regulation ≥ 2) in both panels were highlighted: miR-513a-5p, miR-589-3p, miR-9-5p, miR-135b-5p, miR-506-3p, miR-508-3p, miR-509-3p, miR-518c-5p, miR-663a, and miR-615-3p. Thus, we identified 39 OncomiRs contributing to areca nut-induced carcinogenesis of HNC, with ten being the most significant.

For the tumor-suppressive miRNA (TSmiR) set, we combined the analysis of 156 downregulated miRNAs in ANE sublines and 243 underexpressed miRNAs in tumor tissues. This analysis identified 45 miRNAs (Figure 3C, Table 2), indicating a cluster of TSmiRs downregulated by areca nut exposure, contributing to HNC development through the loss of their malignancy-suppressive function. To identify the most significant TSmiRs, we plotted these miRNAs according to their levels of ANE downregulation and underexpression in tumors. As shown in Figure 3D, eight TSmiRs exhibiting high levels (absolute fold-regulation ≥ 2) in both panels were highlighted: miR-1-3p, miR-410-3p, miR-376c-3p, miR-499a-5p, miR-154-5p, miR-378a-5p, miR-432-5p, and miR-190a-5p. Thus, we identified a signature of 45 TSmiRs affecting the development of areca nut-induced HNC, with eight being the most significant.

### 3.4. Functional Pathways Regulation by Areca Nut-Modulated miRNAs

To investigate how areca nut-modulated miRNAs regulate malignant functions in HNC, we identified target genes of candidate miRNAs and performed pathway analysis on these gene sets. Focusing on oncogenic mechanisms, we examined potential functions regulated by the cluster of eight TSmiR signatures. Given that miRNAs can cross-regulate multiple target genes and vice versa, we employed the multiMiR package algorithm to determine the overall targets. Using the criterion that interactions appear in at least four out of eleven databases in the multiMiR package, we identified 740 genes as targets of the eight TSmiRs (Figure 4A, Appendix A).

The 740-gene set was then analyzed using the KEGG analysis tool. Results shown in Figure 4B reveal that these genes were enriched in ten functional pathways, including molecular signaling pathways such as PI3K-AKT and cellular phenotypes such as focal adhesion and cell cycle. These functional pathways were classified into two phenotypic modules: cell motility and cell survival (Figure 4C).

Among ten pathways, two were related to cell motility (adherens junction and focal adhesion), four were linked to cell survival (EGFR tyrosine kinase inhibitor (TKI) resistance, p53 signaling pathway, cellular senescence, and cell cycle), and the remaining four were associated with both phenotypes (PI3K-AKT signaling, Rap1 signaling, MAPK signaling, and Ras signaling pathways). Thus, we identified that eight TSmiRs cross-regulate multiple pathways related to cell motility and survival, affecting various oncogenic mechanisms in areca nut-induced HNC (Figure 4C, Appendix A).

### 3.5. miR-499a-5p Regulates Multiple Genes Involved in Several Oncogenic Pathways in HNC

Among the eight identified TSmiRs, miR-499a-5p exhibited the most pronounced downregulation following areca nut exposure (Figure 3D), making it a prime candidate for further functional analysis. To investigate the molecular mechanisms potentially regulated by miR-499a-5p, we utilized the multiMiR prediction algorithm to identify its target genes and performed KEGG pathway enrichment analysis to elucidate the regulatory networks involved.

As shown in Figure 5A, a total of 27 genes were predicted to be targeted by miR-499a-5p, mapping to ten oncogenic pathways (Appendix A). Many of these genes are involved in several pathways, and each pathway is regulated by multiple genes. These intricate, cross-linked interactions form a molecular network modulated by miR-499a-5p. For example, VEGF is involved in six oncogenic pathways, including the signaling pathways of Ras, Rap1, MAPK, EGFR, PI3K-Akt, and focal adhesion. Similarly, GSK3β is implicated in four pathways: EGFR signaling, PI3K-Akt signaling, the cell cycle, and focal adhesion. Conversely, the Ras signaling pathway is modulated by various genes, such as ETS1, KRAS, VEGFA, and RAP1B. Likewise, the PI3K-Akt pathway is regulated by multiple genes, including VEGFA, ATF2, GSK3β, and YWHAZ.

Four genes—ETS1, GSK3β, VEGFA, and YWHAZ—which participate in several pathways, were selected for experimental validation. RT-qPCR analysis was conducted to assess the expression of these genes following ectopic transfection of miR-499a-5p in two HNC cell lines (OECM1 and SAS). As shown in Figure 5B, miR-499a-5p suppressed the expression of all four genes in both OECM1 and SAS cells, albeit with varying levels of inhibition. These results confirm that ETS1, GSK3β, VEGFA, and YWHAZ are downstream targets of miR-499a-5p.

To assess the clinical relevance of these four genes and miR-499a-5p in HNC, we analyzed their differential expression between normal and tumor tissues using the TCGA-HNSC dataset. As shown in Figure 5C, ETS1, GSK3β, VEGFA, and YWHAZ were significantly upregulated in tumor tissues (*p* < 0.001), indicating their oncogenic roles in HNC progression. In contrast, miR-499a-5p was markedly downregulated in tumor tissues (*p* < 0.001), further supporting its tumor-suppressive role during HNC development.

### 3.6. Critical Role of miR-499a-5p in Mitigating Areca Nut-Induced Carcinogenesis

Given its pronounced response to areca nut exposure and significant dysregulation in tumor tissues, miR-499a-5p was meticulously examined to confirm its role in areca nut-induced HNC. Arecoline, the predominant active alkaloid in areca nut, was employed to probe this molecular function. RT-qPCR analysis revealed that arecoline inhibited miR-499a-5p expression in various cell types, including normal keratinocytes and HNC cell lines (Figure 6A). This inhibition was dose-dependent, demonstrating a high degree of specificity (Figure 6B).

To further elucidate the effects of miR-499a-5p on cellular functions, we focused on cell motility and survival as these are the primary malignant phenotypes modulated by areca nut-mediated miRNAs (Figure 4C). The Matrigel invasion and wound healing assays were performed to analyze cell motility, while the chemosensitivity assay was used to evaluate cell survival. Two HNC cell lines, OECM1 and SAS, were examined for these functions, and consistent results were observed in both.

Figure 6C shows the results of miR-499a-5p on cell invasion. Overexpression of miR-499a-5p significantly reduced cell invasion. Arecoline facilitated cell invasion, while miR-499a-5p significantly attenuated this arecoline-promoted effect. Figure 6D shows the results of miR-499a-5p on cell migration. Similarly, this miRNA inhibited cell migration and attenuated the arecoline-induced effect. Figure 6E presents the impact of miR-499a-5p on cisplatin sensitivity. Overexpression of miR-499a-5p significantly decreased cell survival. Arecoline increased cisplatin resistance, while miR-499a-5p mitigated the resistance induced by arecoline. Taken together, our results demonstrated that miR-499a-5p is a pivotal regulator of areca nut-induced malignancy, significantly impacting cell invasion, migration, and chemoresistance in HNC.

## 4. Discussion

Areca nut is the primary habitual carcinogen for HNC in Southeast Asia. While miRNAs have emerged as critical molecular players in malignancy, their role in areca nut-induced molecular carcinogenesis remains largely unexplored. This study employed a systematic approach to fill this gap, yielding several key findings (Figure 7). (1) We identified 292 miRNAs induced by areca nut, comprising 136 upregulated and 156 downregulated miRNAs. (2) We discovered 692 miRNAs associated with HNC, including 449 overexpressed and 243 underexpressed in tumor tissues. (3) We defined 39 OncomiRs and 45 TSmiRs influencing the development of areca nut-induced HNC, with ten OncomiR and eight TSmiR signatures being the most significant. (4) We determined that cell motility and survival are the primary malignant phenotypes modulated by areca nut-mediated miRNAs, with multiple oncogenic signaling pathways co-regulating these attributes. (5) MiR-499a-5p emerged as a pivotal TSmiR, significantly impacting cell invasion, migration, chemoresistance, and mitigating areca nut-induced malignant functions in HNC cells. Thus, this study identified a robust panel of miRNA molecules that hold significant potential as biomarkers or therapeutic targets for areca nut-induced HNC. Our findings provide a crucial foundation for future research and therapeutic strategies to combat this prevalent form of cancer.

While our study provides valuable insights, several limitations should be addressed in future research. We identified a set of OncomiRs and TSmiRs associated with areca nut-induced HNC using areca nut-exposed cell lines analyzed through microarray technology. However, this approach may have overlooked potentially significant miRNAs due to the limitations of microarray platforms, which detect only a restricted set of miRNAs. Alternative techniques, such as small RNA sequencing, could be employed to capture a broader range of miRNAs and novel transcripts. Additionally, integrating multi-omics approaches and conducting longitudinal studies could offer a more comprehensive understanding of miRNA dynamics in areca nut-induced carcinogenesis. The complexity of miRNA expression patterns in HNC patients with a history of areca nut chewing warrants further investigation. Future studies should involve larger cohorts of areca nut chewers, considering factors such as the duration and intensity of use and the clinical cancer presentations to better elucidate the direct impact of areca nut on miRNA profiles in human subjects. Clinical association studies that link miRNA profiles with cancer aggressiveness or prognosis could further clarify the potential of specific miRNAs as diagnostic or prognostic biomarkers for areca nut-induced HNC. Addressing these aspects in future research will be crucial for developing more precise prevention strategies and therapeutic interventions for areca nut-associated HNC.

In this study, we identified 39 OncomiRs that contribute to the development of areca nut-induced HNC, with ten of these being the most significant: miR-513a-5p, miR-589-3p, miR-9-5p, miR-135b-5p, miR-506-3p, miR-508-3p, miR-509-3p, miR-518c-5p, miR-663a, and miR-615-3p (Figure 3B). Some of these molecules have been previously implicated in cancer progression, which supports our findings. For instance, miR-9-5p has been reported to promote growth, metastasis, and radioresistance in various cancers, including non-small cell lung cancer (NSCLC), prostate cancer, cervical cancer, and oral cancer [31,32,33,34,35]. Similarly, miR-135b-5p and miR-581c-5p are significantly overexpressed and may serve as potential prognostic biomarkers in HNC [36,37,38]. In breast cancer, miR-615-3p is upregulated and promotes EMT, leading to metastasis [39]. Identifying these miRNAs underscores their potential roles in the malignancy associated with areca nut exposure. However, miR-513a-5p, miR-589-3p, miR-506-3p, miR-508-3p, miR-509-3p, and miR-663a have not been extensively studied in the context of HNC or areca nut exposure. These miRNAs represent novel findings in our work, highlighting their potential as new biomarkers or therapeutic targets.

We also identified a signature of 45 TSmiRs impacting the development of areca nut-induced HNC, with the eight most significant being miR-1-3p, miR-410-3p, miR-376c-3p, miR-499a-5p, miR-154-5p, miR-378a-5p, miR-432-5p, and miR-190a-5p (Figure 3D). Most of these miRNAs have been reported to play tumor-suppressive roles in various cancers. For instance, miR-376c-3p is reduced in gastric cancer, breast cancer, and HNC, contributing to tumor progression [40,41,42]. MiR-154-5p acts as a tumor suppressor and potential prognostic biomarker in several cancers, including osteosarcoma, esophageal squamous cell carcinoma, and HNC [43,44,45,46]. MiR-378a-5p exhibits tumor-suppressive properties in breast cancer, hepatocellular carcinoma, colorectal cancer, and HNC [47,48,49,50,51]. MiR-432-5p was also found downregulated in osteosarcoma, colorectal cancer, and HNC [52,53,54,55]. Additionally, miR-1-3p and miR-499a-5p are underexpressed in HNC [56,57]. Notably, miR-1-3p and miR-190-5p were also reported to be downregulated in smoke-induced lung cancer, consistent with our findings of reduced expression of these TSmiRs in areca nut-exposed HNC [58]. These results suggest that exposure to carcinogens such as areca nut can lead to the loss of stress-protective or malignancy-suppressive functions of TSmiRs, thereby contributing to cancer development. Identifying these TsmiRs underscores their potential as a therapeutic intervention strategy in areca nut-induced HNC.

It is worth noting that different miRNAs have been reported in various HNC studies conducted by other research groups. Examples are the TSmiRs miR-17-5p, miR-22, miR-329, miR-486-3p, and miR-866-3p downregulated by arecoline and modulated malignant phenotypes in HNC cell lines [59,60,61,62,63]. Other relevant studies indicate some miRNAs (miR-21, miR-23a, miR-10b, miR-29b, and miR-200b/c) participate in areca nut-associated oral fibroblastic activity [20,64,65,66,67,68]. This variability in findings can be attributed to factors such as biological heterogeneity, differences in study methodology, and technical variability. Understanding these factors is crucial for interpreting the variability in miRNA findings across different HNC studies. Despite these differences, commonalities in miRNA signatures can provide valuable insights into the underlying mechanisms of HNC and highlight potential biomarkers and therapeutic targets.

Understanding the specific miRNAs and their target pathways provides valuable insights into the molecular mechanisms underlying areca nut-induced carcinogenesis. To investigate the malignant phenotype regulated by miRNAs, we conducted enrichment analysis using the eight TSmiR signatures, which revealed pathways associated with cell motility and survival (Figure 4). This dysregulation underscores the complex interplay between areca nut exposure, miRNA regulation, and tumor progression in HNC. Our findings are consistent with previous studies showing that areca nut affects several key signaling pathways, including the p53 signaling pathway, MAPK signaling pathway, PI3K-AKT signaling pathway, and Ras signaling pathway. These pathways mediate EMT and oxidative stress, promoting cell motility and survival, while these aberrant activations are common features in many cancers, including HNC [69,70,71,72,73]. Our enrichment analysis highlights that areca nut-induced dysregulation of these eight TSmiR signatures disrupts the normal regulatory mechanisms of these pathways, thereby enhancing malignant phenotypes such as increased cell motility and survival.

In this study, miR-499a-5p, a critical miRNA showing significant downregulation in clinical tumor tissues and reduced by areca nut exposure, was demonstrated to play a pivotal role (Figure 5 and Figure 6). Previously, miR-499a-5p has been reported to be downregulated in various cancers, exhibiting tumor-suppressive functions. In NSCLC, this molecule is reduced in expression in tumors and predicts a poor prognosis [74]. In breast cancer, miR-499a-5p is downregulated and has been shown to suppress the ferroptosis pathway [75]. In endometrial cancer, exosomal miR-499a-5p acts as a tumor suppressor by inhibiting metastasis [76]. In cervical cancer, miR-499a-5p inhibits EMT and enhances radiosensitivity [77]. Notably, miR-499a-5p is downregulated in oral fibroblasts stimulated by areca nut [78,79]. Our study further supports these findings, demonstrating that ectopic expression of miR-499a-5p significantly suppresses cell migration and invasion and increases cisplatin sensitivity in HNC cell lines. Moreover, miR-499a-5p attenuated the effects induced by arecoline, including enhanced cell motility and chemoresistance. The consistent downregulation of miR-499a-5p across various cancers and its tumor-suppressive functions underscore its critical role in cancer biology. In HNC specifically, miR-499a-5p appears to be a key regulator of malignant phenotypes influenced by areca nut exposure. Targeting miR-499a-5p could serve as a novel strategy to mitigate the adverse effects of areca nut exposure and enhance the efficacy of existing cancer therapies.

## 5. Conclusions

This study identifies a panel of miRNAs involved in areca nut-induced HNC. We discovered 39 OncomiRs and 45 TSmiRs, highlighting their roles in cell motility, survival, and therapeutic resistance. Key pathways such as p53, MAPK, PI3K-AKT, and Ras were significantly impacted. A critical miRNA, miR-499a-5p, was shown to regulate cancer progression, with its restoration suppressing cell migration, invasion, and chemoresistance. These findings highlight the potential of miRNA-based biomarkers and therapeutic targets in combating areca nut-induced HNC. This research provides a foundation for novel therapeutic strategies and underscores the need for further validation and clinical application in precision medicine.

## Figures and Tables

**Figure 1 cancers-16-03710-f001:**
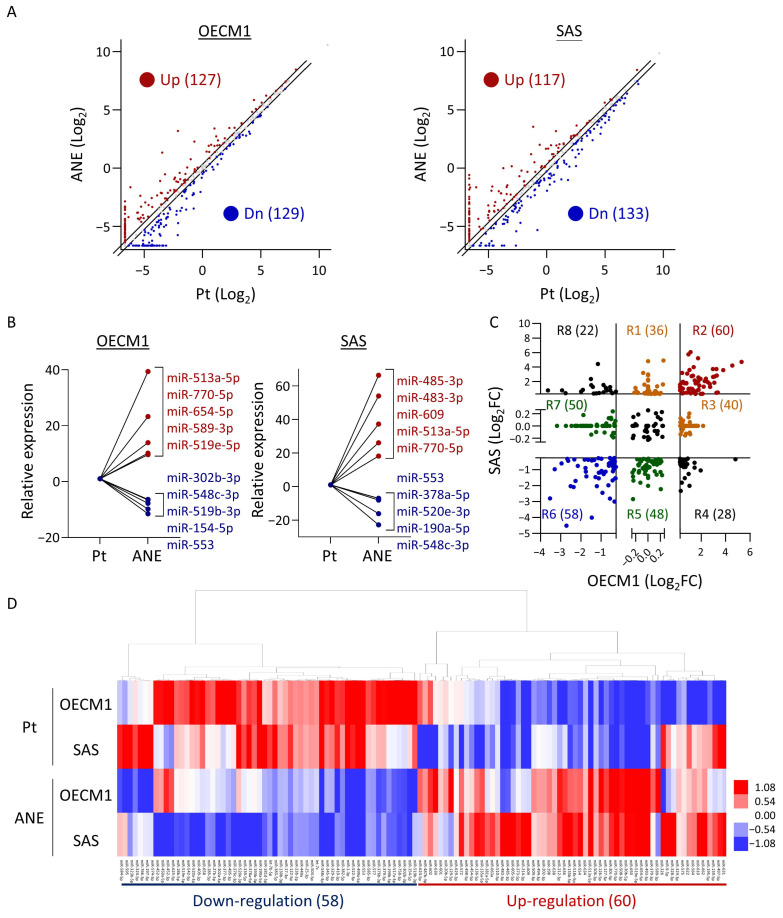
Areca nut induces changes in miRNA expression profiles. (**A**) Scatter plot showing upregulated and downregulated miRNAs in OECM1 and SAS cell lines after ANE treatment (absolute fold-regulation ≥ 1.2). Red and blue dots represent upregulated and downregulated miRNAs, respectively. (**B**) Top five upregulated and downregulated miRNAs in OECM1 and SAS cells. (**C**) Scatter plot comparing ANE-induced miRNA expression changes between OECM1 and SAS cells. Dot colors represent different regions (R1–R8). (**D**) Clustergram of 60 upregulated and 58 downregulated miRNAs (from regions R2 + R6 in panel (**C**)) under chronic ANE exposure. Pt: parental. ANE: areca nut extract.

**Figure 2 cancers-16-03710-f002:**
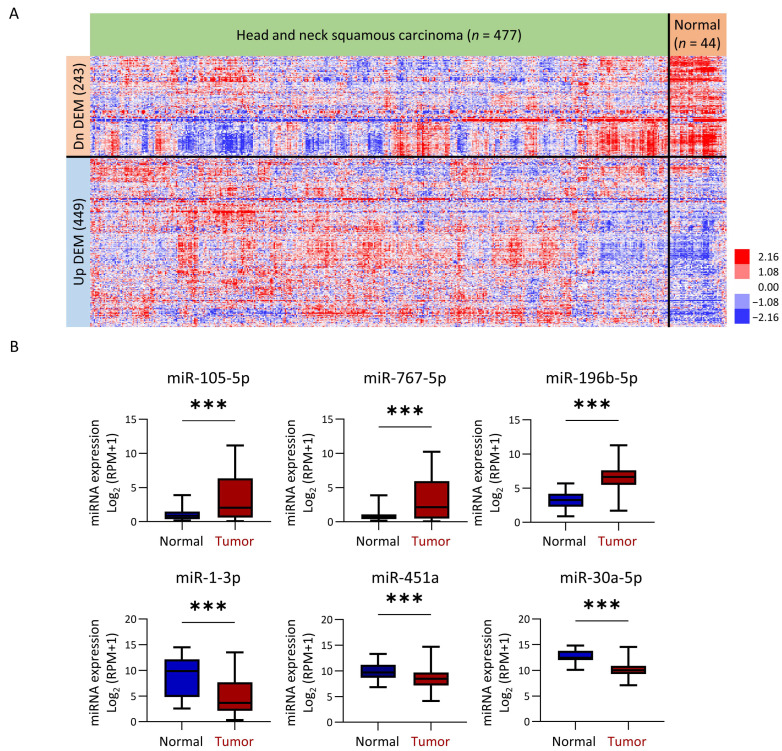
Differential miRNA expression profile in TCGA-HNSC. (**A**) Heatmap showing 449 upregulated and 243 downregulated miRNAs in tumor tissues compared to adjacent normal tissues. (**B**) Expression of three upregulated and three downregulated miRNAs in clinical HNC patients. (***, *p* < 0.001, *t*-test).

**Figure 3 cancers-16-03710-f003:**
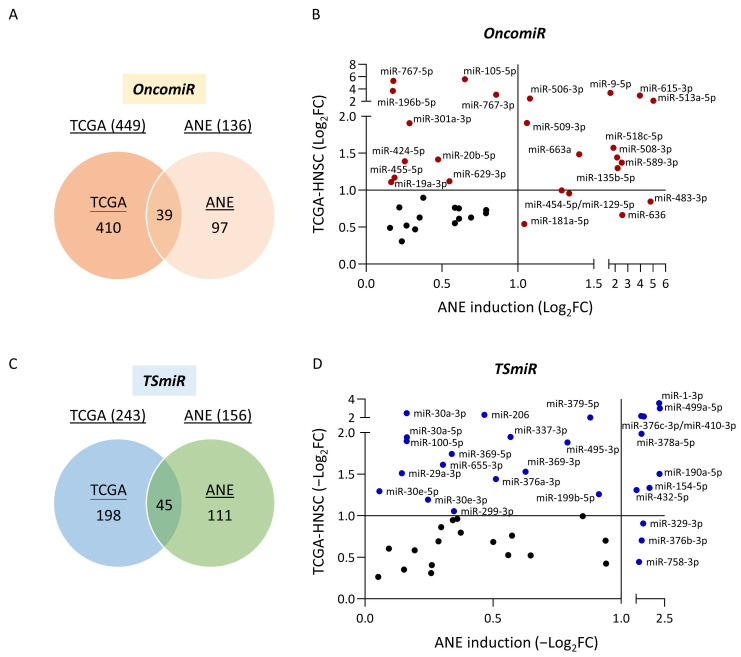
Integration of ANE-dysregulated miRNAs and mis-expressed miRNAs in HNC patients. (**A**,**C**) Venn diagrams showing the overlap between ANE-dysregulated miRNAs and mis-expressed miRNAs in HNC patients, identifying oncogenic miRNAs (OncomiRs) and tumor suppressor miRNAs (TSmiRs). (**B**,**D**) Scatter plots of OncomiRs and TSmiRs. X-axis: fold change of ANE induction; Y-axis: fold change in TCGA-HNSC. Criteria: absolute fold change ≥ 2 in both groups. Ten OncomiR and eight TSmiR signatures are highlighted.

**Figure 4 cancers-16-03710-f004:**
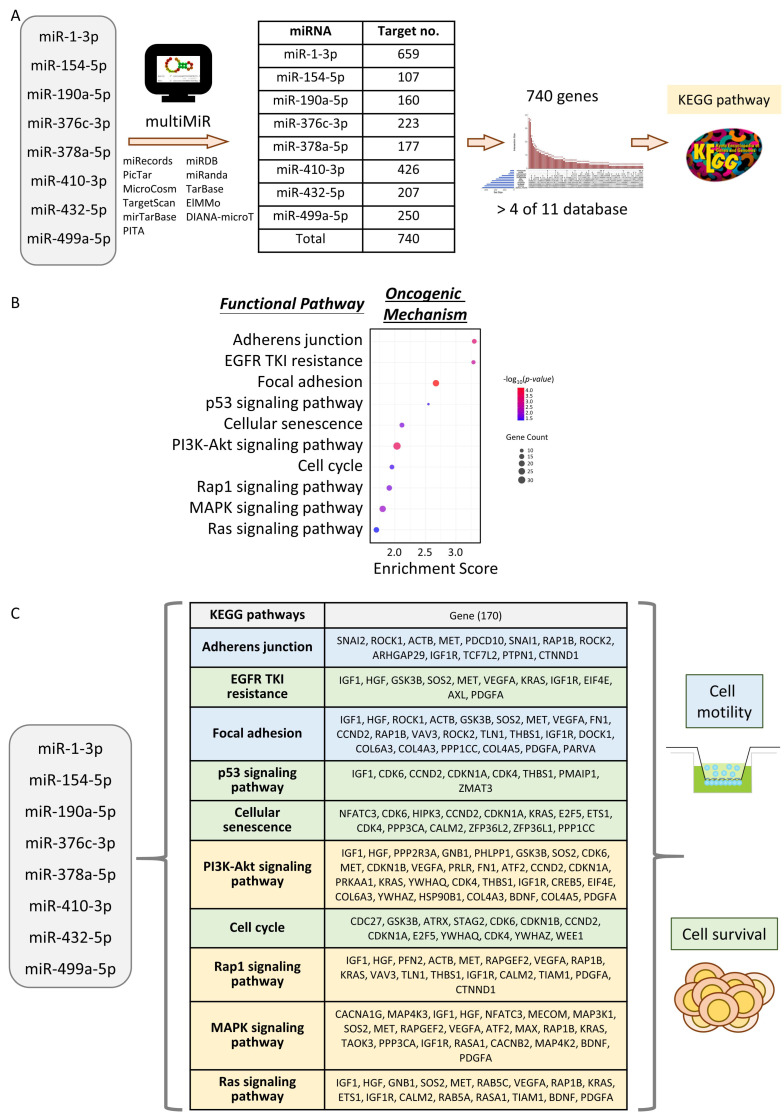
Oncogenic functions of eight TSmiR signatures. (**A**) Flowchart illustrating the process of investigating the functional roles of eight TSmiR signatures. (**B**) Enrichment analysis of eight TSmiR signatures in ten oncogenic mechanisms. (**C**) Overview of eight TSmiR signatures in ten oncogenic pathways and their potential target genes involved in cell motility and survival.

**Figure 5 cancers-16-03710-f005:**
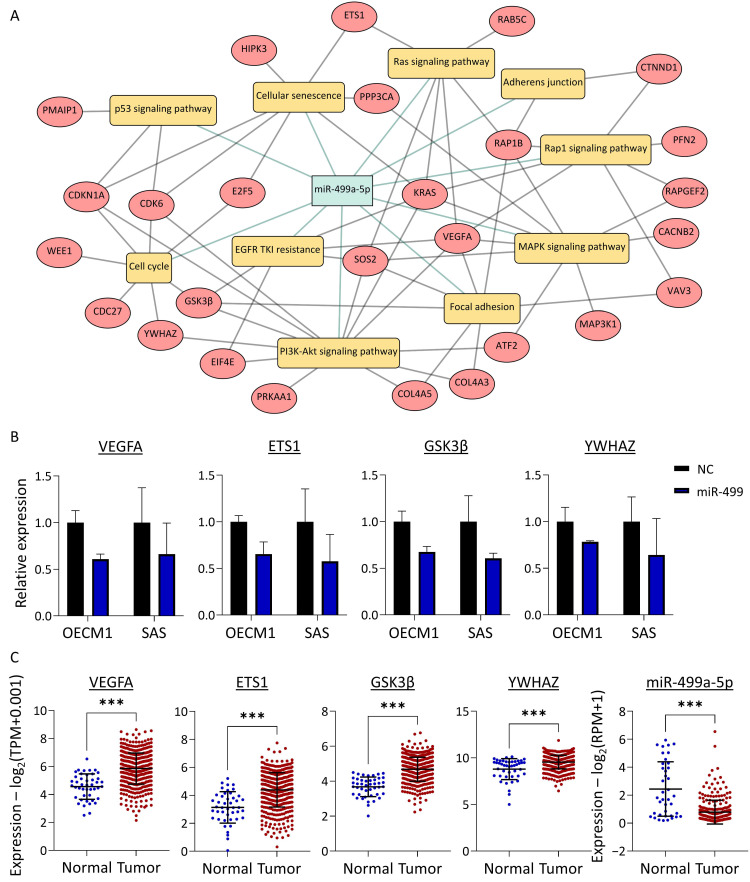
Target gene analysis and validation of miR-499a-5p in HNC. (**A**) Interaction network showing the predicted target genes of miR-499a-5p and their involvement in oncogenic pathways. (**B**) Validation of miR-499a-5p’s regulatory effects on target genes ETS1, GSK3β, VEGFA, and YWHAZ with or without miR-499a-5p overexpression. (**C**) Expression of ETS1, GSK3β, VEGFA, and YWHAZ in clinical TCGA-HNSC dataset. (***, *p* < 0.001, *t*-test).

**Figure 6 cancers-16-03710-f006:**
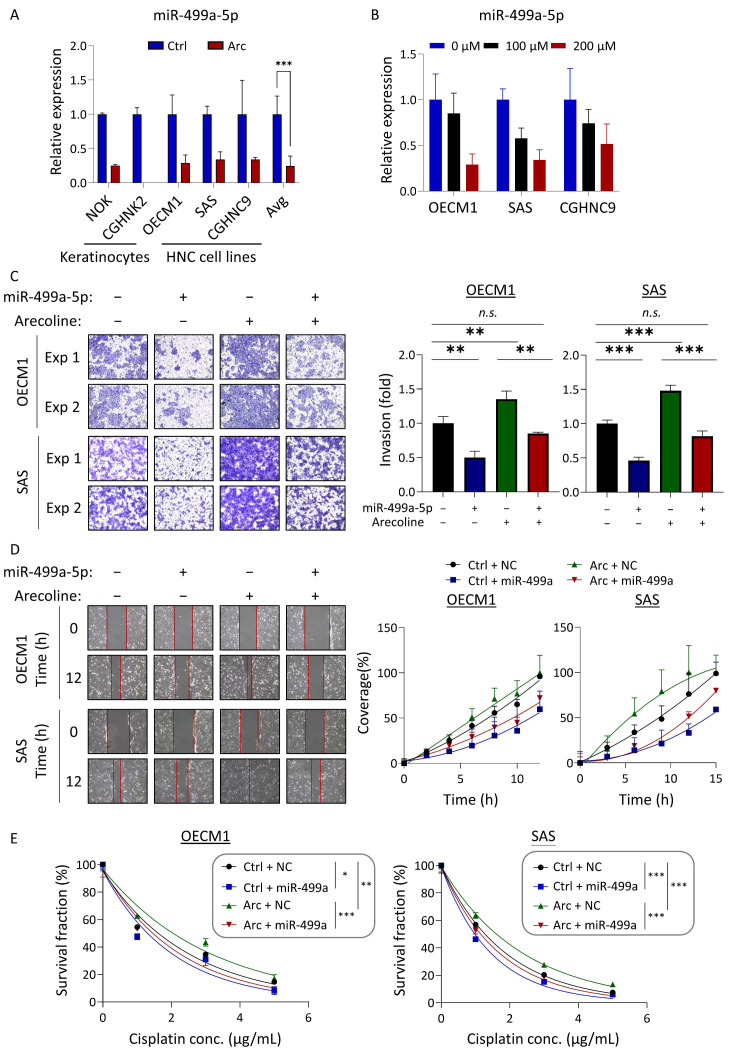
Areca nut enhances cell motility and survival by suppressing miR-499a-5p expression. (**A**) miR-499a-5p expression in two keratinocytes and three HNC cell lines treated with arecoline. (**B**) Dose-dependent effects of arecoline (0–200 µM) on miR-499a-5p expression in OECM1, SAS, and CGHNC9 cells. (**C**–**E**) Assessment of cell invasion (**C**), migration (**D**), and chemosensitivity (**E**) in HNC cells under certain conditions: miR-499a-5p overexpression, arecoline treatment, and the combined miR-499a-5p overexpression and arecoline treatment. (*, *p* < 0.05; **, *p* < 0.01; ***, *p* < 0.001, *n.s.*, non-significance, *t*-test and ANOVA).

**Figure 7 cancers-16-03710-f007:**
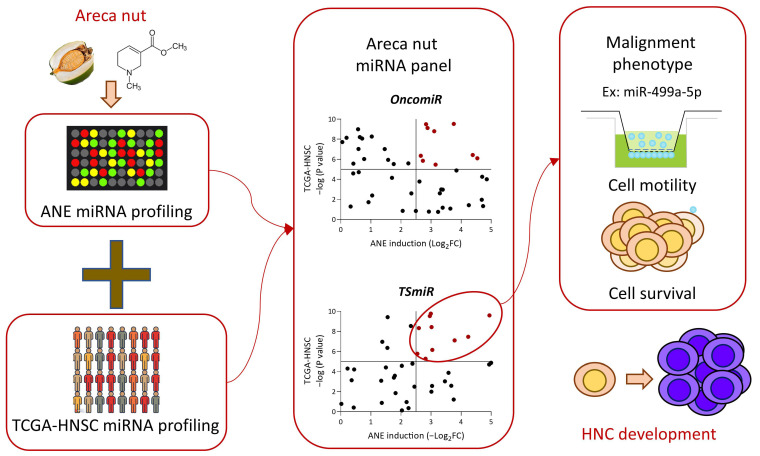
Summary model of areca nut-induced miRNA profiling in HNC.

**Table 1 cancers-16-03710-t001:** List of OncomiRs upregulated by areca nut and overexpressed in tumor tissues.

miRNA	TCGA-HNSC		ANE Induced Panel		
	T/N	*p*	OEC FC	SAS FC	Average FC
miR-513a-5p	4.28	1.90 × 10^−3^	39.40	26.15	32.77
miR-483-3p	1.80	8.71 × 10^−4^	1.94	54.03	27.99
miR-615-3p	7.56	9.83 × 10^−24^	1.19	29.95	15.57
miR-636	1.58	4.40 × 10^−4^	8.35	3.54	5.95
miR-589-3p	2.59	5.05 × 10^−14^	10.19	1.39	5.79
miR-135b-5p	2.46	1.31 × 10^−7^	3.41	5.82	4.62
miR-508-3p	2.72	2.85 × 10^−2^	4.88	4.11	4.49
miR-518c-5p	2.98	2.03 × 10^−2^	3.36	3.96	3.66
miR-9-5p	10.26	3.01 × 10^−14^	2.91	3.30	3.10
miR-663a	2.80	8.42 × 10^−6^	1.43	3.87	2.65
miR-129-5p	1.94	1.04 × 10^−2^	3.32	1.73	2.53
miR-454-5p	2.00	6.92 × 10^−11^	1.53	3.35	2.44
miR-506-3p	5.41	7.73 × 10^−3^	2.21	2.02	2.11
miR-509-3p	3.76	4.58 × 10^−3^	2.58	1.59	2.08
miR-181a-5p	1.46	3.75 × 10^−9^	1.21	2.91	2.06
miR-767-3p	8.32	1.26 × 10^−20^	2.62	1.00	1.81
miR-324-3p	1.66	5.75 × 10^−12^	2.20	1.26	1.73
miR-576-5p	1.61	1.14 × 10^−7^	2.46	1.00	1.73
miR-33b-5p	1.55	4.37 × 10^−3^	2.23	1.00	1.62
miR-105-5p	48.05	1.52 × 10^−16^	2.14	1.00	1.57
miR-501-5p	1.53	4.17 × 10^−4^	2.15	0.91	1.53
miR-142-3p	1.69	6.95 × 10^−5^	1.95	1.10	1.53
miR-33a-5p	1.70	5.59 × 10^−4^	0.87	2.13	1.50
miR-342-3p	1.47	4.75 × 10^−3^	2.07	0.93	1.50
miR-629-3p	2.18	7.85 × 10^−16^	1.20	1.72	1.46
miR-20b-5p	2.67	1.04 × 10^−2^	1.47	1.31	1.39
miR-222-3p	1.86	8.44 × 10^−17^	1.68	0.92	1.30
miR-551a	1.55	1.03 × 10^−2^	1.00	1.55	1.28
miR-191-5p	1.38	7.19 × 10^−5^	1.60	0.90	1.25
miR-301a-3p	3.75	4.28 × 10^−27^	1.11	1.33	1.22
miR-590-5p	1.44	3.50 × 10^−6^	0.98	1.42	1.20
miR-424-5p	2.62	4.37 × 10^−13^	1.11	1.28	1.19
miR-181a-3p	1.24	5.97 × 10^−3^	1.14	1.21	1.18
miR-20a-5p	1.70	3.39 × 10^−5^	1.24	1.09	1.16
miR-455-5p	2.25	5.93 × 10^−21^	1.23	1.04	1.14
miR-767-5p	39.86	7.03 × 10^−19^	1.26	1.00	1.13
miR-196b-5p	12.84	4.35 × 10^−36^	0.96	1.30	1.13
miR-19a-3p	2.16	4.85 × 10^−9^	1.04	1.21	1.12
miR-141-3p	1.40	1.42 × 10^−3^	0.86	1.37	1.11

FC: fold-change. T/N: tumors versus normal tissues.

**Table 2 cancers-16-03710-t002:** List of TSmiRs downregulated by areca nut and underexpressed in tumor tissues.

miRNA	TCGA-HNSC		ANE Induced Panel		
	T/N	*p*	OEC FC	SAS FC	Average FC
miR-499a-5p	−7.97	5.04 × 10^−4^	−5.68	−4.08	−4.75
miR-190a-5p	−2.83	2.83 × 10^−8^	−2.75	−16.09	−4.69
miR-1-3p	−11.77	3.44 × 10^−4^	−5.12	−4.25	−4.64
miR-154-5p	−2.52	1.61 × 10^−6^	−9.82	−1.96	−3.26
miR-410-3p	−4.36	4.31 × 10^−4^	−3.37	−2.18	−2.65
miR-329-3p	−1.87	4.52 × 10^−4^	−2.43	−2.76	−2.59
miR-376b-3p	−1.63	1.29 × 10^−2^	−4.45	−1.69	−2.45
miR-378a-5p	−3.96	7.84 × 10^−5^	−1.48	−7.02	−2.44
miR-376c-3p	−4.48	7.18 × 10^−4^	−3.43	−1.86	−2.41
miR-758-3p	−1.36	2.13 × 10^−3^	−6.19	−1.36	−2.23
miR-432-5p	−2.47	2.09 × 10^−5^	−1.18	−7.20	−2.03
miR-409-3p	−1.34	1.15 × 10^−3^	−1.71	−2.18	−1.92
miR-382-5p	−1.62	6.75 × 10^−4^	−2.33	−1.63	−1.92
miR-199b-5p	−2.39	3.64 × 10^−8^	−3.52	−1.29	−1.88
miR-379-5p	−4.05	7.97 × 10^−5^	−3.13	−1.30	−1.84
miR-377-3p	−1.99	3.27 × 10^−4^	−2.75	−1.34	−1.80
miR-495-3p	−3.69	4.36 × 10^−4^	−2.53	−1.31	−1.73
let-7b-5p	−1.43	1.27 × 10^−6^	−1.30	−1.97	−1.57
miR-369-3p	−2.89	5.21 × 10^−5^	−3.39	−1.00	−1.54
miR-548b-3p	−1.69	3.11 × 10^−2^	−2.91	−1.00	−1.49
miR-337-3p	−3.86	3.99 × 10^−4^	−1.00	−2.87	−1.48
miR-485-5p	−1.44	3.54 × 10^−2^	−2.80	−1.00	−1.47
miR-376a-3p	−2.71	5.16 × 10^−3^	−3.68	−0.88	−1.43
miR-23b-3p	−1.61	1.90 × 10^−9^	−1.40	−1.43	−1.41
miR-206	−4.94	8.71 × 10^−4^	−2.31	−0.99	−1.38
miR-128-3p	−1.74	4.36 × 10^−3^	−1.02	−1.78	−1.30
miR-27b-3p	−1.95	1.15 × 10^−9^	−1.32	−1.25	−1.28
miR-299-3p	−2.08	4.51 × 10^−3^	−1.75	−1.00	−1.27
miR-491-5p	−1.93	4.62 × 10^−4^	−0.98	−1.79	−1.27
miR-369-5p	−3.35	2.58 × 10^−6^	−1.72	−1.00	−1.26
miR-655-3p	−3.06	1.59 × 10^−3^	−1.61	−1.00	−1.24
miR-215-5p	−1.82	4.93 × 10^−6^	−1.67	−0.97	−1.23
miR-140-5p	−1.62	5.66 × 10^−8^	−1.32	−1.14	−1.22
miR-199a-5p	−1.32	1.82 × 10^−2^	−1.49	−1.00	−1.20
miR-107	−1.24	2.25 × 10^−3^	−1.13	−1.27	−1.20
miR-30e-3p	−2.29	2.42 × 10^−11^	−1.31	−1.08	−1.19
miR-152-3p	−1.50	2.91 × 10^−6^	−1.05	−1.26	−1.14
miR-30a-5p	−3.85	1.76 × 10^−9^	−1.42	−0.93	−1.12
miR-30a-3p	−5.55	1.15 × 10^−8^	−1.42	−0.92	−1.12
miR-100-5p	−3.72	1.02 × 10^−16^	−0.91	−1.45	−1.12
miR-181c-5p	−1.28	8.72 × 10^−4^	−1.42	−0.91	−1.11
miR-29a-3p	−2.85	4.19 × 10^−14^	−1.00	−1.24	−1.11
let-7f-5p	−1.52	7.75 × 10^−6^	−1.22	−0.94	−1.07
miR-30e-5p	−2.45	5.16 × 10^−11^	−1.34	−0.85	−1.04
miR-532-5p	−1.20	3.58 × 10^−3^	−0.87	−1.28	−1.04

FC: fold-change. T/N: tumors versus normal tissues.

## Data Availability

The data of TCGA-HNSC used in this study are from the UCSC Xena platform (https://xena.ucsc.edu/, accessed on 10 May 2024).

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
