# Peer review of "MiRNA Profiling of Areca Nut-Induced Carcinogenesis in Head and Neck Cancer"

_cancers, 2024, doi:10.3390/cancers16213710_

Round 1

Reviewer 1 Report

Comments and Suggestions for Authors

In this manuscript, Huang et. al., identify a panel of miRNAs involved in areca nut-induced head and neck cancer. Using integrative analysis, they found miR-499a-5p as a potential tumor suppressor in cancer progression by controlling cell migration, invasion, and chemoresistance.  

Overall, the manuscript is well organized. The rationale is clearly defined, the experiments sound well-controlled and designed to prove the hypothesis. There are only a few improvements that should be performed before the acceptance.

1- Figure D is too small - it is impossible to read the names of the miRNAs. The authors should provide a list in the supplementary files.

2- It would be good to add a molecular network correlating miR-499a-5p and its target genes found in Figure 4C.

3- To make a strong case about the role of miR-499a-5p, the authors should explore the gene expression of at least one target of miR-499a-5p based on data from Figure 4, such as members of the PI3K-AKT pathway.

Author Response

In this manuscript, Huang et. al., identify a panel of miRNAs involved in areca nut-induced head and neck cancer. Using integrative analysis, they found miR-499a-5p as a potential tumor suppressor in cancer progression by controlling cell migration, invasion, and chemoresistance.  

Overall, the manuscript is well organized. The rationale is clearly defined, the experiments sound well-controlled and designed to prove the hypothesis. There are only a few improvements that should be performed before the acceptance.

Comment 1: Figure 1D is too small - it is impossible to read the names of the miRNAs. The authors should provide a list in the supplementary files.

Response 1: Thank you for your valuable feedback. We agree that the small size of Figure 1D makes it difficult to read the miRNA names. To resolve this, we have included a detailed list of all miRNAs presented in Figure 1D in the newly added Supplementary Table S4. We believe this addition will enhance the accessibility and clarity of the data while preserving the visual summary in the main text.

Comment 2: It would be good to add a molecular network correlating miR-499a-5p and its target genes found in Figure 4C.

Response 2: We appreciate this insightful suggestion. In response, we have added a new results section (Figure 5) to explore the molecular network of miR-499a-5p and its target genes. Figure 5A presents a network visualization, highlighting the predicted targets of miR-499a-5p and mapping these targets to their associated KEGG pathways. Additionally, in Figure 5B, we performed RT-qPCR analysis to validate four target genes modulated by miR-499a-5p in two HNC cell lines, supporting their involvement in various signaling pathways.

Comment 3: To make a strong case about the role of miR-499a-5p, the authors should explore the gene expression of at least one target of miR-499a-5p based on data from Figure 4, such as members of the PI3K-AKT pathway.

Response 3: We appreciate this insightful suggestion. In response, we have added a new results section (Figure 5) to explore the molecular network of miR-499a-5p and its target genes. Figure 5A presents a network visualization, highlighting the predicted targets of miR-499a-5p and mapping these targets to their associated KEGG pathways. Additionally, in Figure 5B, we performed RT-qPCR analysis to validate four target genes modulated by miR-499a-5p in two HNC cell lines, supporting their involvement in various signaling pathways.

Reviewer 2 Report

Comments and Suggestions for Authors

Areca nut (betel nut) chewing is crucial for Asian head and neck cancer (HNC) pathogenesis. In detail, areca nut extract and arecoline, a major alkaloid component, have been demonstrated to induce the pathogenesis of oral cancer and oral submucous fibrosis. The presented study aimed to systematically identify a miRNA panel associated with areca nut-induced HNC.

The authors found 39 oncogenic miRNAs (OncomiRs) and 45 tumor-suppressive miRNAs (TsmiRs) imported into HNC development in Asia. Importantly, miR-499a-5p was shown to regulate cancer progression, with its restoration suppressing cell migration, invasion, and chemoresistance. Thus, miRNAs can act as biomarkers and therapeutic targets in the field of areca nut-induced HNC. 

The study is well-designed and the manuscript is properly written. Data shown can potentially have meaning in the future in the therapeutic approach to HNC. However, before publication, the manuscript needs several minor corrections:

A) The abstract is too long (above 200 words); please keep the word limit.

B) Table 2 - the content of this table should not be bolded.

Author Response

Areca nut (betel nut) chewing is crucial for Asian head and neck cancer (HNC) pathogenesis. In detail, areca nut extract and arecoline, a major alkaloid component, have been demonstrated to induce the pathogenesis of oral cancer and oral submucous fibrosis. The presented study aimed to systematically identify a miRNA panel associated with areca nut-induced HNC.

The authors found 39 oncogenic miRNAs (OncomiRs) and 45 tumor-suppressive miRNAs (TsmiRs) imported into HNC development in Asia. Importantly, miR-499a-5p was shown to regulate cancer progression, with its restoration suppressing cell migration, invasion, and chemoresistance. Thus, miRNAs can act as biomarkers and therapeutic targets in the field of areca nut-induced HNC. 

The study is well-designed and the manuscript is properly written. Data shown can potentially have meaning in the future in the therapeutic approach to HNC. However, before publication, the manuscript needs several minor corrections:

Comment 1: The abstract is too long (above 200 words); please keep the word limit.

Response 1: Thank you for your attention to the abstract length. According to the journal’s guidelines, the word limit for the abstract is 250 words, and our original version contained 226 words, which was within this requirement. However, we appreciate the reviewer’s suggestion and have revised the abstract to be more concise while maintaining the essential content and findings. The updated abstract is now 210 words, ensuring clarity and readability without losing critical information about the study.

Comment 2: Table 2 - the content of this table should not be bolded.

Response 2: Thank you for your careful observation. We have corrected this formatting issue by removing the bold styling from all content in Table 2. The table now follows standard formatting guidelines, enhancing its readability and ensuring consistency with the rest of the manuscript.

Reviewer 3 Report

Comments and Suggestions for Authors

In this manuscript titled ,, MiRNA profiling of Areca Nut-Induced Carcinogenesis in Head and Neck Cancer”, the authors analyze the MiRNA profile in an in vitro study of induction of carcinogenesis in neck and head cancer (HNC) by betel nut. The main aim of the study was to evaluate the involvement of miRNA in areca nut induced HNC. The topic is original and relevant to the field. It addresses a specific gap in the field covering the panel of miRNAs involved in areca nut-induced HNC. Compared with other published material the authors describe in detail the current methods of studying. The conclusions support the results and the discussions because the authors demonstrated through the complex and sophisticated statistical methods used the areca nut exposure miRNA profiling and panel and also, they highlighted a malignant phenotype. There are almost 80 references, very recent from the topic of the study carried out. The figures, tables and the supplementary material complete and highlight the explanations better.

The report has some minor issues and I have some suggestions to potentially improve your manuscript:

In the introduction, you refer to "HNC" as head and neck cancer, which is formally incorrect. In the context of the betel nut induced cancer, betel nut causes oral and tongue cancer. The term "head and neck cancer" also includes extraoral tumors. I recommend to give a briefly defining at the beginning of manuscript.

Did the study have the approval of an ethics committee?

One or two paragraphs about study limitations and possible new research directions are needed.

Otherwise, great manuscript.

Author Response

In this manuscript titled, MiRNA profiling of Areca Nut-Induced Carcinogenesis in Head and Neck Cancer”, the authors analyze the MiRNA profile in an in vitro study of induction of carcinogenesis in neck and head cancer (HNC) by betel nut. The main aim of the study was to evaluate the involvement of miRNA in areca nut induced HNC. The topic is original and relevant to the field. It addresses a specific gap in the field covering the panel of miRNAs involved in areca nut-induced HNC. Compared with other published material the authors describe in detail the current methods of studying. The conclusions support the results and the discussions because the authors demonstrated through the complex and sophisticated statistical methods used the areca nut exposure miRNA profiling and panel and also, they highlighted a malignant phenotype. There are almost 80 references, very recent from the topic of the study carried out. The figures, tables and the supplementary material complete and highlight the explanations better.

The report has some minor issues and I have some suggestions to potentially improve your manuscript:

Comment 1: In the introduction, you refer to "HNC" as head and neck cancer, which is formally incorrect. In the context of the betel nut induced cancer, betel nut causes oral and tongue cancer. The term "head and neck cancer" also includes extraoral tumors. I recommend to give a briefly defining at the beginning of manuscript.

Response 1: Thank you for your valuable suggestion. While areca nut is most commonly associated with oral cancer, research indicates that its carcinogenic effects may extend to extraoral regions such as the larynx and pharynx, which are also at increased risk. For this reason, we prefer to use the term “head and neck cancer (HNC)” to reflect the broader scope of its potential impact. We have revised reference 8 to support this point and have defined the anatomic sites of HNC at the beginning of the manuscript. The revised text is as follows: “Head and neck cancer (HNC) primarily comprises squamous cell carcinomas affecting the oral cavity, pharynx, larynx, and salivary glands, ranking among the top ten most prevalent cancers worldwide.” We appreciate the reviewer’s attention to this detail.

Comment 2: Did the study have the approval of an ethics committee?

Response 2: Thank you for raising this important point regarding ethical considerations. Our study exclusively utilized established cell lines and did not involve any clinical samples. The human subject data used in this study were sourced from the publicly available TCGA-HNSC dataset via the UCSC Cancer Genome Browser. This information is clearly stated in the Methods section (point 2.8). In accordance with our institution’s guidelines and standard research practices, studies that use only commercially available cell lines or publicly accessible datasets do not typically require ethics committee approval. We appreciate the reviewer’s concern and hope this clarifies our approach.

Comment 3: One or two paragraphs about study limitations and possible new research directions are needed.

Response 3: Thank you for the valuable suggestion. In response, we have added a second paragraph in the discussion section to address the potential limitations of our study and outline future research directions. Specifically, we highlighted the primary limitation: the restricted number of miRNAs detectable by microarray technology. We also proposed several critical directions for future research, including the use of more advanced RNA sequencing techniques to capture a broader range of miRNAs and conducting studies with larger clinical cohorts to validate our findings for potential clinical applications. These additions present a balanced reflection of our study's limitations while outlining actionable directions for future research, offering a roadmap for advancing this field.